# The Effect of Snow Depth on Spring Wildfires on the Hulunbuir from 2001–2018 Based on MODIS

**Hong Ying [1], Yu Shan [1,2], Hongyan Zhang [1,*], Tao Yuan [3], Wu Rihan [1] and Guorong Deng [1]**

[1]    School of Geographical Sciences, Northeast Normal University, Changchun 130024, China; hongy864@nenu.edu.cn (H.Y.); yushangis@163.com (Y.S.); wurh651@nenu.edu.cn (W.R.); denggr272@nenu.edu.cn (G.D.)

[2]    Inner Mongolia Key Laboratory of Remote Sensing and Geographic Information System, Hohhot 010022, China

[3]    Geological Exploration Fund Management Center of Jilin Province, Changchun 130061, China; yttdgh@126.com

*    Correspondence: zhy@nenu.edu.cn; Tel.: +86-431-850-99550

**Abstract:** Wildfires are one of the important disturbance factors in natural ecosystems and occur frequently around the world. Detailed research on the impact of wildfires is crucial not only for the development of livestock husbandry but also for the sustainable use of natural resources. In this study, based on the Moderate Resolution Imaging Spectroradiometer (MODIS) burned area product MC464A1 and site snow depth measurements, the kernel density estimation method (KDE), unary linear regression analysis, Sen + Mann-Kendall trend analysis, correlation analysis, and R/S analysis were used to evaluate the relationship between snow and spring wildfires (SWFs) in Hulunbuir. Our results indicated that SWFs decreased during the period of 2001–2018, were mainly distributed in the eastern portion of the study area, and that the highest SWF density was 7 events/km$^2$. In contrast, the maximum snow depth increased during the period of 2001–2018 and the snow depth was deeper in the middle but shallower in the east and west. The SWFs and snow depth have significant negative correlations over space and time. The snow depth mainly affects the occurrence of SWFs indirectly by affecting the land surface temperature (LST) and Land Surface Water Index (LSWI) in spring. The snow depth was positively correlated with the LSWI in most of Hulunbuir and strongly negatively correlated with the LST, and this correlation was stronger in the eastern and western regions of Hulunbuir. The results of the Hurst exponent indicated that in the future, the snow depth trend will be opposite that of the current state, meaning that the trend of decreasing snow depth will increase dramatically in most of the study area, and SWFs may become more prominent. According to the validation results, the Hurst exponent is a reliable method for predicting the snow depth tendency. This research can be based on the snow conditions of the previous year to identify areas where fires are most likely to occur, enabling an improved and more targeted preparation for spring fire prevention. Additionally, the present study expands the theory and methods of wildfire occurrence research and promotes research on disasters and disaster chains.

**Keywords:** wildfire; snow; correlation analysis; Hulunbuir

## 1. Introduction

Wildfires represent all fires that occur in natural ecosystems [1] and have important effects on vegetation dynamics, the biogeochemical cycles of carbon, nitrogen, and other elements, atmospheric chemistry, and the climate [2]. They contribute to atmospheric pollution as well. Wildfires have considerable negative impacts on livestock husbandry production and grassland ecological environments [3–5]. They are one of the most important disturbance factors in natural

ecosystems and occur frequently around the world. The occurrence of wildfires not only seriously restricts the development of animal husbandry but also poses a serious threat to the living environment and homeland security of humans. An average of 348 Mha were burned annually from 1997 to 2011 around the world, although the global burned area showed a decreasing trend at a reduction rate of 4.3 Mha yr$^{-1}$ during the period from 2000–2012 [6].

The Inner Mongolia Autonomous Region of China (IMC) is the main wildfire area among the arid and semi-arid regions of China, and grassland fires, forest fires, and cropland fires [7] occur frequently in the region every year; the main fire sources are human activities [8]. These wildfires mainly occur in spring and autumn each year and account for more than 90% of the fires annually [9,10]. Hulunbuir is the easternmost prefecture-level city in IMC, and it is rich in grassland and forest resources. Statistical analyses show that from 1981 to 1999, a total of 892 grassland fires occurred in the eastern part of the IMC for an average of 46.95 fires per year, and the most frequent occurrence was in Hulunbuir [11,12]. Hulunbuir has less precipitation in spring and autumn, dry weather, and a high number of strong wind days, and most of the grassland and forest areas are relatively sparsely populated and have low traffic. These factors make it very difficult to manage fires in this region [12]. Therefore, detailed research on the impact of wildfires is crucial, not only for the development of livestock husbandry but also for the sustainable use of natural resources.

Advances in the field of remote sensing have greatly increased the potential for fire monitoring [13]. Many researchers have used large-scale observations of remote sensing and short-term characteristics to obtain substantial data [14–16]. In combination with spatial statistics and geographic information technologies, spatial point process methods—such as Ripley's K function, nearest neighbor indication, kernel density, Poisson models, and correlation analysis—have been used in studies on the spatial and temporal distribution of large-scale fires. Furthermore, satellite products have become the key source of global information on fire occurrence because they provide comprehensive spatial and temporal coverage of fire-affected areas [13]. Various satellite sensors have been used to evaluate fire occurrence, including the Advanced Very-High-Resolution Radiometer (AVHRR) [17,18] and the Along-Track Scanning Radiometer (ATSR) [19]. Additionally, observations have been collected based on imagery from the Visible and Infrared Scanner (VIRS) onboard the Tropical Rainfall Measuring Mission (TRMM) [20], GOES geostationary satellites [21], Meteosat Second Generation (MSG) [22,23], Himawari-8 [24–26], and Landsat [27]. The Moderate-Resolution Imaging Spectroradiometer (MODIS) is a large-scale remote sensing monitoring instrument developed by the National Aeronautics and Space Administration (NASA) in the late 1990s, and the Terra and Aqua satellites have been equipped with sensors since 18 December 1999 and 4 May 2002, respectively. For the detection of fire burned areas, the spatial resolution of the MODIS sensors is 500 m and measurements are recorded twice per day (10:30 AM and 1:30 PM during the day and 10:30 PM and 1:30 AM at night for Terra and Aqua, respectively) [13]. MODIS is the most widely used sensor for active fire detection, and MODIS data have been used in a broad range of fire-related applications, such as the estimation of gas emissions from fires and the analysis of fire regimes [28], the distribution of fire on different land cover types [29], and the probability of fire occurrence in future climate scenarios [30,31]. Forest fires have been better studied than grassland fires, and the point pattern analysis method was applied to forestry statistics in 2000 [32]. In the temperate rain forest of Canada, the probability of a grid cell burning was positively correlated with the summer temperature and negatively correlated with precipitation and distance to municipalities, campgrounds, dirt roads, railroads, and paved roads [33]. At the county level in Florida, population and poverty were positively correlated with the annual wildfire area and intensity-weighted fire area [34]. The results of the application of spatial statistical analysis in Canada showed that fire locations were spatially clustered in Ontario [35]. The spatial distribution pattern of grassland fires in Hulunbuir has also gradually attracted attention [36–38]. Moreover, several studies have documented a relationship between snow and temperature or spring precipitation [39] and noted that temperature and precipitation are important climate factors that affect wildfire occurrence [40].

However, most research on wildfires has concentrated on wildfire hazards, disaster risk assessments, and early warning, and few studies have analyzed the factors that influence the occurrence of wildfires [8,10,41,42]. Furthermore, the climate data used in these studies are observation data on the timing of wildfires, and although these data were provided by the Meteorological Bureau, they do not consider the changes in surface temperature and surface humidity that are caused by the periodic or aperiodic changes in climatic elements, such as snow. Therefore, the combination of rapidly developing technologies such as remote sensing and geographic information system data is required to extract and analyze various information, such as wildfire occurrence and snow depth, and the impact of the snow distribution on wildfires must be investigated to provide updated methods and technical support for wildland disaster management. The main objective of this study was to quantify the effect of snow on spring wildfires. To achieve this objective, we used the MODIS burned area data product MCD64A1 and land cover data to extract wildfires from 2001 to 2018, and we integrated the snow data to reveal the relationship between snow and wildfires in Hulunbuir. In addition, we analyzed the influence of winter snow on the main climatic factors of wildfires that occurred during the following year, i.e., the effect of surface temperature and surface humidity on wildfires. We also predicted the future trend of snow depth. This study aimed to provide a quantitative perspective on the effect of snow on wildfires and strengthen the ability to predict and provide early warnings of wildfires. This research also has great practical value for reducing the losses caused by fire disasters.

## 2. Materials and Methods

### 2.1. Study Area

The study area is located in Hulunbuir (prefecture-level city) in the IMC (115.22–126.06°E, 47.08–53.23°N), covers an area of approximately 252.948 km$^2$, and is comprised of 13 counties: (a) Erguna, (b) Genhe, (c) Oroqen, (d) Hailar, (e) Arun, (f) Morin Dawa, (g) Oldbarag, (h) Yakeshi, (i) Zhalantun, (j) Xinbarag left, (k) Xinbarag right, (l) Manjvvr, and (m) Ewenki. Heilongjiang Province borders the study area to the east, Russia to the north, and Mongolia to the west. Elevations range from 167–1675 m, and the topography is high in the center but low in both the east and the west (Figure 1). The study area experiences a typical temperate continental climate with low annual precipitation, frequent droughts, and windy periods during the winter and spring seasons. The mean annual temperature is approximately −20 °C in winter and approximately 20 °C in summer, and the annual precipitation is 320 mm. More than 55% of the annual precipitation is concentrated in the summer [12]. The vegetation types in the Hulunbuir grassland area consist of a variety of plant communities. The main vegetation communities are the *Stipa baicalensis* community, the *Filifolium sibiricum* community, and the *Leymus chinensis* community.

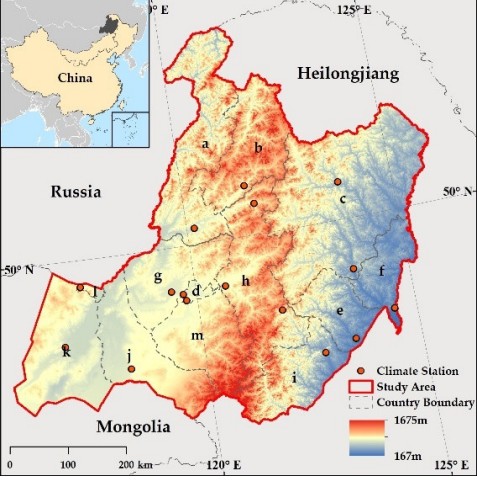

**Figure 1.** Location of the study area.

*2.2. Data*

2.2.1. Spring Wildfire Dataset

The spring (March, April, May) wildfire (SWF) dataset covers 18 years (2001–2018) from the Collection 6 MODIS Burned Area product MCD64A1. The MCD64A1 Burned Area Product is a monthly Level-3 gridded 500-m product. Data were obtained from NASA's Level-1 and Atmosphere Archive & Distribution System Distributed Active Archive Center (LAADS DAAC, https://ladsweb.modaps.eosdis.nasa.gov/). The study area is covered by four images, i.e., h25v03, h25v04, h26v03, and h26v04. The original data from MCD64A1 included 5 types of information comprised of the burn date, burn date uncertainty, QA, first day, and last day [43]. In this study, we mainly used the burn date layer data, in which the individual pixels were assigned to one of 25 classes. The meaning of each class is shown in Table 1.

**Table 1.** MOD14/MYD14 fire mask pixel classes.

| Class | Meaning |
|---|---|
| −2 | Water |
| −1 | Unmapped due to insufficient data |
| 0 | Unburned land |
| 1–366 | Ordinal day of burn |

The MODIS Reprojection Tool (MRT) was used to reformat the downloaded datasets from the Earth Observing System (EOS). The MRT is a mosaic, resampling, and reprojection tool commonly used for multiple images. In this paper, to avoid errors in the area calculation, we only set the mosaic after extracting the burn date band via the MRT; we did not use the projection transformation and resampling. In addition, we converted the extracted raster burn data to feature points.

2.2.2. Snow Depth Data

The monthly maximum snow depth data for winter (December, the following January, February) at 118 meteorological stations distributed throughout the IMC (Hulunbuir includes 16 climate stations, and the distribution is shown in Figure 1) during the period of 1979–2018 were acquired from the Meteorological Bureau of the IMC, and they represent observations of the maximum snow depth in one month per station. We calculated the average snow depth for three months per station and defined this value as the average snow depth for a year. Based on the average annual snow depth for each station, we obtained the continuous distribution (spatial resolution is 1 km × 1 km) of the average snow depth in each year throughout the IMC using ANUSPLIN 4.3, which is a special software package for surface fitting of climate data. This software is based on spline interpolation theory, and it allows the introduction of multivariate and covariate linear submodels, and the model coefficients can be determined automatically according to the data. Thus, ANUSPLIN can process two or more splines smoothly, which allows the introduction of multiple impact factors as covariates for spatial interpolation of meteorological elements. More importantly, it can perform spatial interpolation of multiple surfaces at the same time, which is especially suitable for meteorological time series data [44–46]. ANUSPLIN has been widely used internationally. In this study, we used a digital elevation model (DEM) as a covariate to interpolate the snow depth. Finally, the gridded snow depth for winter with a spatial resolution of 1 km × 1 km over the Hulunbuir region spanning the period of 2001–2018 were extracted using mask analysis.

2.2.3. LSWI Data

MODIS has 3 bands that are sensitive to the moisture water content: $NIR_1$ (1230–1250 nm), $SWIR_1$ (1628–1652 nm), and $SWIR_2$ (2105–2155 nm) [47]. In this paper, the Land Surface Water Index (LSWI) in spring (3–5 months) was estimated by the ratio of two bands with a lower absorption rate and a higher

water absorption rate. The two bands were obtained from MOD09A1 (8-day, 0.5 km), and the MRT was used to mosaic and reproject the data and screen the shortwave infrared (SWIR) and near-infrared (NIR) bands. The LSWI was calculated as follows:

$$LSWI = \frac{NIR - SWIR_1}{NIR + SWIR_1},$$ (1)

where *LSWI* is the mean Land Surface Water Index in spring, and $NIR_1$ and $SWIR_1$ are the values of the near-infrared and the shortwave infrared bands, respectively.

### 2.2.4. LST Data

This paper applies MOD11A2 (8-day, 1-km, https://lpdaac.usgs.gov/) data to characterize the land surface temperature (LST) in spring. The first band of MOD11A2 was extracted using the MRT, and it was reprojected and resampled. We converted the original values to degrees Celsius (original value $\times 0.02 - 273.15$).

### 2.2.5. Land Cover Data

To more accurately extract the distribution of wildfires, we used the MODIS Land Cover Type Product (MCD12Q1) to remove nonwildfire areas. These data were obtained by the Earth Observing System Data and Information System (EOSDIS, https://earthdata.nasa.gov), which supplies global maps of land cover at annual time steps and 500-m spatial resolution for 2001–2017. We used the MRT to extract the first band (International Geosphere-Biosphere Programme) of MCD12Q1 and remove urban and built-up lands, permanent snow and ice, and water bodies. Wildland includes information as shown in Table 2. We used the annual wildland data to screen the annual wildfires using the Extract by Mask tool in ArcGIS (the 2018 wildfire data were applied to 2017 years of wildland for extraction).

**Table 2.** The land cover type of Hulunbuir extracted from MODIS Land Cover Type Product (MCD12Q1).

| Value | Name | Value | Name |
|:---:|:---:|:---:|:---:|
| 1 | Evergreen Needleleaf Forests | 9 | Savannas |
| 3 | Deciduous Needleleaf Forests | 10 | Grasslands |
| 4 | Deciduous Broadleaf Forests | 11 | Permanent Wetlands |
| 5 | Mixed Forests | 12 | Croplands |
| 7 | Open Shrublands | 14 | Cropland/Natural Vegetation Mosaic |
| 8 | Woody Savannas | 16 | Barren |

*2.3. Methods*

### 2.3.1. Density of SWF Points

Kernel density estimation (KDE) is a method used to reconstruct the probability density function from random sampling points. KDE does not have any prior density assumptions, and it provides a high-quality probability density estimation when given a suitable bandwidth [48]. The KDE method supports only point and linear features. The SWF in this study is a randomly occurring point event, and this method was used to address the spatial density distribution of SWF points [10]. The associated formula is as follows:

$$f(x) = \sum_{i=1}^{n} \frac{1}{d^2} k \left[ \frac{(x - x_i)}{d} \right],$$ (2)

where *x* represents any point in space, and $x \ldots x_i$ is the spatial location of n SWF points. Additionally, *k* () is a bivariate probability density function called the core, *d* is the bandwidth used to define the size of the smoothing, and the radius of a circle is centered on *x*. In this equation, the spatial distribution

density of SWFs can be generated for different scales by adjusting the $d$. After many tests using the KDE analysis, we found it reasonable to set the $d$ to 22 km [12].

### 2.3.2. Trend Analysis

(1) The interannual variability and trends of snow depth and wildfires

A time series linear regression model based on the least squares method can be used to simulate the trend in time series data. We used linear regression analysis to more intuitively analyze the interannual fluctuations and trends of SWF and snow depth. The slope of the trend is obtained by the least squares method, and the formula is:

$$slope = \frac{n \times \sum_{i=1}^{n}(i \times y) - \sum_{i=1}^{n} i \sum_{i=1}^{y} y}{n \times \sum_{i=1}^{n} i^2 - \left(\sum_{i=1}^{n} i\right)^2}, \tag{3}$$

where *slope* is the trend, and $y$ and $i$ represent the values of the time series and years, respectively.

(2) Spatial trend analysis

The Theil-Sen median trend analysis and the Mann-Kendall test method can be combined and become an important method for determining the trend of long-term sequence data. These methods have been gradually applied to long-term sequence analysis. Theil-Sen median trend analysis is a nonparametric statistical trend calculation method that can reduce the impact of data outliers [49]. We chose this method to prevent abnormal values from affecting the spatial variation trend of snow depth, LSWI, and LST. The Theil-Sen median trend calculates the median of the slope of $n(n-1)/2$ data combinations and is calculated as follows:

$$\beta = median\left(\frac{x_j - x_i}{j - i}\right), \tag{4}$$

where $x_j$ and $x_i$ represent the sequence value at time $j$ and $i$, respectively; when $\beta > 0$, snow depth or SWF occurrence increases (and vice versa).

Mann-Kendall is a nonparametric statistical test method used to determine the significance of a trend. This test does not require a sample to obey a certain distribution and is not affected by a few outliers. The test process is as follows:

$$S = \sum_{i=1}^{n-1} \sum_{j=i+1}^{n} sign(x_j - x_i), \text{ and } sign = \begin{cases} 1 & x_j - x_i > 0 \\ 0 & x_j - x_i = 0 \\ -1 & x_j - x_i < 0 \end{cases}, \tag{5}$$

where $x_j$ and $x_i$ are the sequential data values, $n$ is the length of the series, and *sign* is a symbolic function. The Mann-Kendall's statistic $Z_{MK}$ is calculated as follows:

$$Z_{MK} = \begin{cases} \frac{S-1}{\sqrt[2]{Var(S)}} & S > 0 \\ 0 & S = 0 \\ \frac{S+1}{\sqrt[2]{Var(S)}} & S < 0 \end{cases} \tag{6}$$

$$Var(S) = \frac{n(n-1)(2n+5)}{18} \tag{7}$$

At a given significance level $a$, when $|Z| > U_{1-\alpha/2}$, there is a significant change in the sequence at the $\alpha$ level.

### 2.3.3. Correlation Analysis between SWFs and Influencing Factors

When analyzing the relationship between two variables in a given space, we need to know whether there is a close quantitative relationship between the two variables. The statistical indicator

used to explain the close relationship between the variables $(x, y)$ of sample size $n$ is called the correlation coefficient, denoted by $R$, and the formula for calculating $R$ is as follows:

$$R = \frac{\sum_{i=1}^{n} (x_i - \overline{x})(y_i - \overline{y})}{\sqrt{\sum_{i=1}^{n} (x_i - \overline{x})^2 \sum_{i=1}^{n} (y_i - \overline{y})^2}}, \tag{8}$$

where $\overline{x}$ and $\overline{y}$ are the average of the variables $x$ and $y$, respectively, and the correlation coefficient $R$ has a range $[-1,1]$. For $R > 0$, there is a positive correlation between the two variables. For $R < 0$, there is a negative linear correlation between the two variables. For $R = 0$, there is no linear relationship between the two variables.

### 2.3.4. Hurst Exponent and R/S Analysis

R/S analysis is a basic method for nonlinear time series analysis. The so-called R/S analysis is actually a rescaled range analysis. Given a time series, we calculate the range of series representing the growth rate or the rate of decay; then, we calculate the range ($R$) and standard deviation ($S$) corresponding to different time lags and find the ratio of the two (R/S). If R/S lags at any time and presents a power law distribution, the power exponent is the so-called Hurst exponent. Based on this, we can estimate the evolution of the trend of a time series as a function of time [50,51]. Using this method, we predict the difference (consistent) between the future trend of snow depth and the current trend. The specific process is as follows. Consider a time series increment $\{\xi(t)\}$, where $\xi(t) = B(t) - B(t-1)$ and $B(t)$ is the observed value at time $t$ ($t = 1, 2, \dots$). For any positive $\tau$, we define the mean series as follows:

$$\langle \xi \rangle_t = \frac{1}{\tau} \sum_{t=1}^{\tau} \xi(t), \tag{9}$$

where $\tau = 1, 2, 3, \dots$, represents the time lag. Here, $X(t)$ is used to represent the cumulative dispersion:

$$X(t, \tau) = \sum_{t=1}^{\tau} (\xi(t) - \langle \xi \rangle_t), \tag{10}$$

where $1 \leq t \leq \tau$. Then, the range $R(\tau)$ is defined as follows:

$$R(\tau) = \max_{1 \leq t \leq \tau} [X(t, \tau)] - \min_{1 \leq t \leq \tau} [X(t, \tau)], \tag{11}$$

The standard deviation $S(\tau)$ is defined by:

$$S(\tau) = \left\{ \frac{1}{\tau} \sum_{t=1}^{\tau} [(\xi(t) - \langle \xi \rangle_t)]^2 \right\}^{\frac{1}{2}} \tag{12}$$

Based on $R(\tau)$ and $S(\tau)$, we obtain the following:

$$\frac{R(\tau)}{S(\tau)} \hat{=} \frac{R}{S} \tag{13}$$

After long-term theoretical analysis and simulation experiments, Hurst found the following empirical relationship:

$$\frac{R(\tau)}{S(\tau)} \propto \left( \frac{\tau}{2} \right)^H, \tag{14}$$

where $H$ is the Hurst exponent andindicates proportionality. For $H = 1/2$, the time series exhibit no changes. For $H > 1/2$, the changes in the time series are positively correlated with prior changes. This implies that the time series is persistent; an increase in the past means an increment in the future, and a reduction in the past means a reduction in the future. For $H < 1/2$, the changes in the time series are negatively correlated with prior changes. This condition implies that the time series is anti-persistent. Thus, an increase in the past means a reduction in the future and a reduction in the past that means an increase in the future.

## 3. Results

### 3.1. Relationships between SWFs and Snow Depth

#### 3.1.1. Change and Distribution of SWFs

According to the statistical analysis of MCD64A1, there was a different SWF area in the study region each year. To clarify the trend in SWF area, we conducted a trend analysis.

From the interannual variations and results of the trend analysis of the SWF area during the period of 2001–2018 (Figure 2), the SWF area in Hulunbuir over the last 18 years was the largest (7384.01 km$^2$) in 2008, followed by 2003. The SWF area was 6894.89 km$^2$, and the fluctuation in the SWF area was small in the other years and had an overall decreasing trend, although this was not significant. Furthermore, snow depth fluctuated between 180.95 km$^2$ and 2546.57 km$^2$.

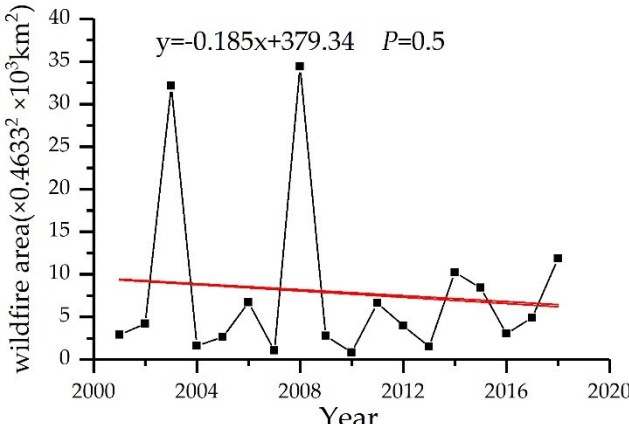

**Figure 2.** Interannual variation in spring wildfires (SWFs) area during the period of 2001–2018.

During the period of 2001–2018 in the study area, the total SWF area reached 30,032.75 km$^2$ according to the MCD64A1 data. The distribution and density of SWF events are shown in Figure 3. SWFs were mostly distributed in the eastern and northwestern parts of the study area, such as areas c, f, i, h, and e, and accounted for 36.9%, 23.4%, 10.7%, 8.85%, and 8.74% of the total SWF areas, respectively. The spatial distribution of SWF density revealed that the highest density was located in areas c and f, reaching 7 events/km$^2$.

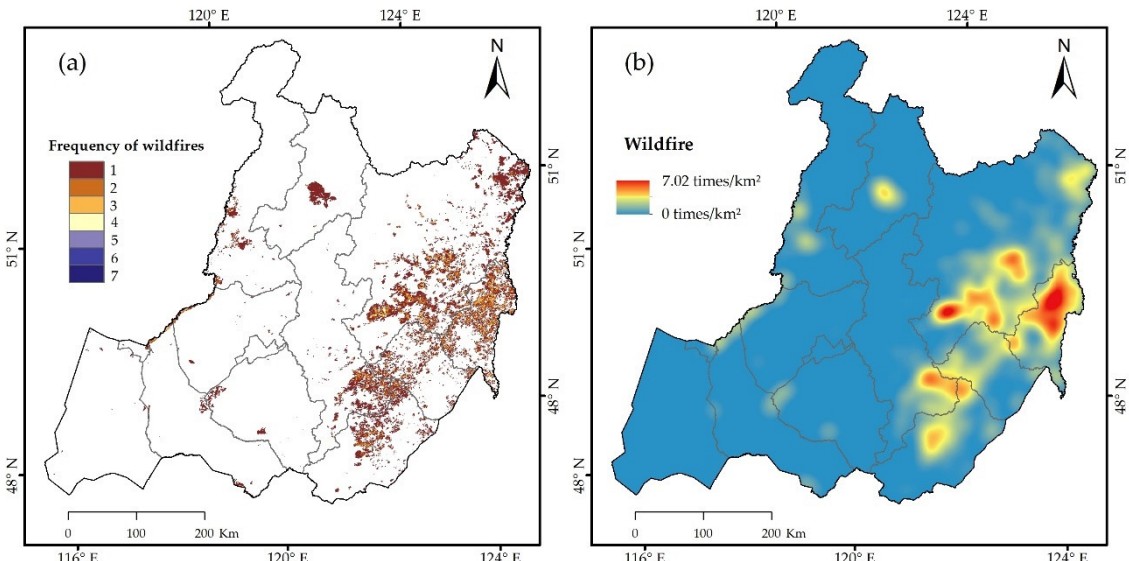

**Figure 3.** The distribution frequency (**a**) and density (**b**) of SWFs during the period of 2001–2018.

### 3.1.2. Temporal and Spatial Variation in Snow Depth during the Period of 2001–2018

According to the results of the interannual changes and trends in the mean snow depth, the mean snow depth increased between 2001 and 2018 (Figure 4). The snow depth data from the 16 stations in the study area were interpolated to obtain the spatial distribution of the mean snow depth during the period of 2001–2018, as shown in Figure 5a. The snow depth was high in the center of the study area and low in both the east and the west; the deepest snow was 28 cm, and the lowest snow depth was 6 cm.

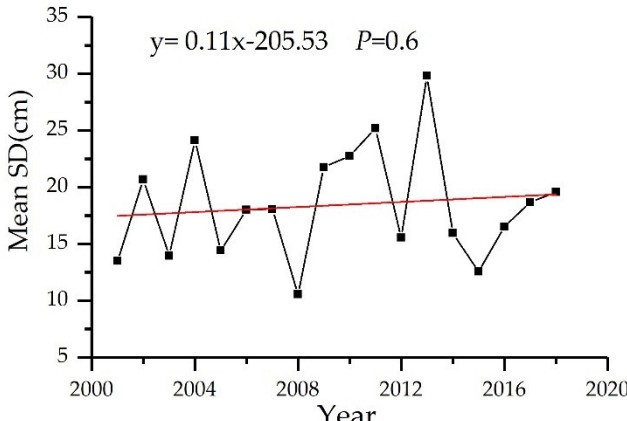

**Figure 4.** Interannual variation in mean of snow depth during the period of 2001–2018.

Figure 5b shows the spatial variation in snow depth for Hulunbuir during the period of 2001–2018. The figure shows that the increasing and decreasing trends in snow depth accounted for 54.2% and 45.8% of the study area, respectively. The areas showing a decreasing trend were mainly distributed in the north, central, and northwest regions of the study area, while the increasing trend area was mainly confined to the eastern and western parts of the study area.

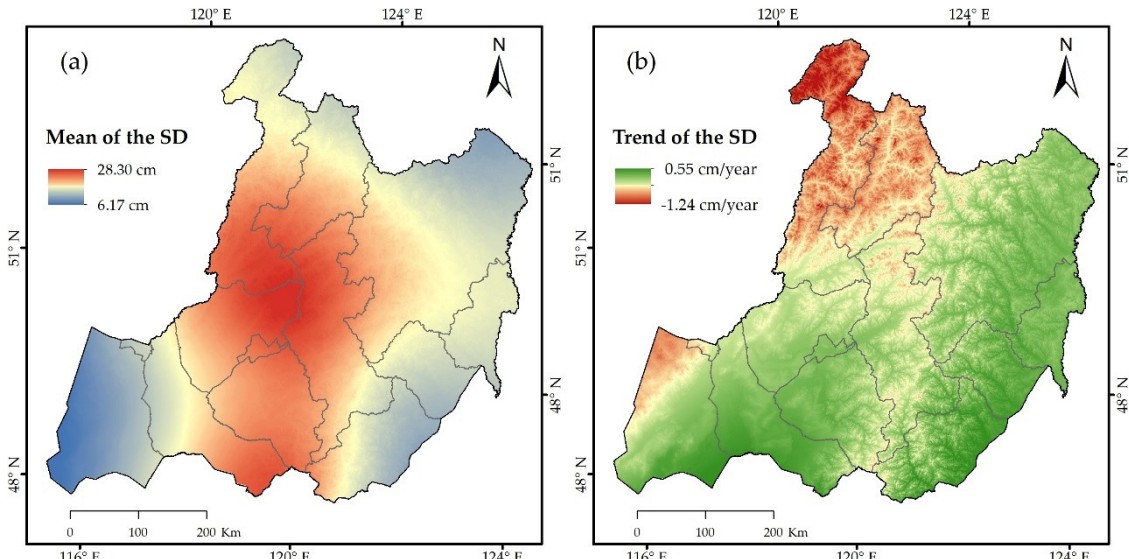

**Figure 5.** Distribution of (**a**) mean of snow depth and (**b**) trend of annual snow depth in 2001–2018.

### 3.1.3. Correlation between SWFs and Snow Depth

The correlation coefficient was used to evaluate the correlation between SWFs and snow depth in the study area for the past 18 years. The results are shown in Table 3.

**Table 3.** Correlation between SWFs and snow depth during the period of 2001–2018.

| Variable | Pearson Correlation | Significance Test | N |
| --- | --- | --- | --- |
| Spatial density of SWFs and spatial mean snow depth | −0.17 | 0.01 | 252,287 |
| Annual total SWF area and annual mean snow depth | −0.516 | 0.05 | 18 |

According to the correlation analysis results of the spatial density of SWFs and the mean snow depth distribution data after interpolation in the study area over the past 18 years, the correlation between the two was negative and the correlation coefficient was −0.17, which passed the significance test at the 0.01 level. The negative correlation coefficients of total SWFs and snow depth for each year at the 16 meteorological stations were as low as −0.516 and passed the significance test at the 0.05 level.

### 3.2. The Mechanistic Effect of Snow on Wildfires

SWFs are affected by many external factors, such as climate conditions, environmental aspects, and fuel. Climate factors have an important influence on the temporal and spatial distribution patterns of SWFs because these factors change the tendency, regularity, and distribution of fires. In the study area, the maximum snow depth generally appears in January. As the temperature increases gradually beginning in February each year, the snow enters the thawing period, and the coverage of snow depth decreases gradually. Snow melting affects the temperature of the surface, and the combustibles and soil moisture also change. Temperature and humidity are the main climatic factors affecting the occurrence of SWFs. Therefore, snow depth indirectly affects the occurrence of SWFs. In this paper, we selected the LSWI, which characterizes the surface moisture condition, and the LST, which characterizes the surface temperature condition, to analyze the indirect effect of snow depth on SWF.

### 3.2.1. Spatial Distribution and Trends of the LSWI and LST in Spring

The distribution of the mean LSWI exhibited great spatial heterogeneity in the study area during the period of 2001–2018, with a mean LSWI of 0–0.99 in spring (Figure 6a). The LSWI was larger



in the west of the study area and smaller in the east and north, with areas d, g, and m having the largest values and area i the smallest. Figure 6b shows the spatial variation trend in the LSWI during the period of 2001–2018; 43.7% of the LSWI values showed an increasing trend (2.62% of the pixels showed significant increases), and these were mainly distributed in the eastern and western parts of the study area, while 56.3% of the areas showed a decreasing trend (9.2% of the pixels showed significant increases, which were mainly distributed in north and certain areas in the center of Hulunbuir).

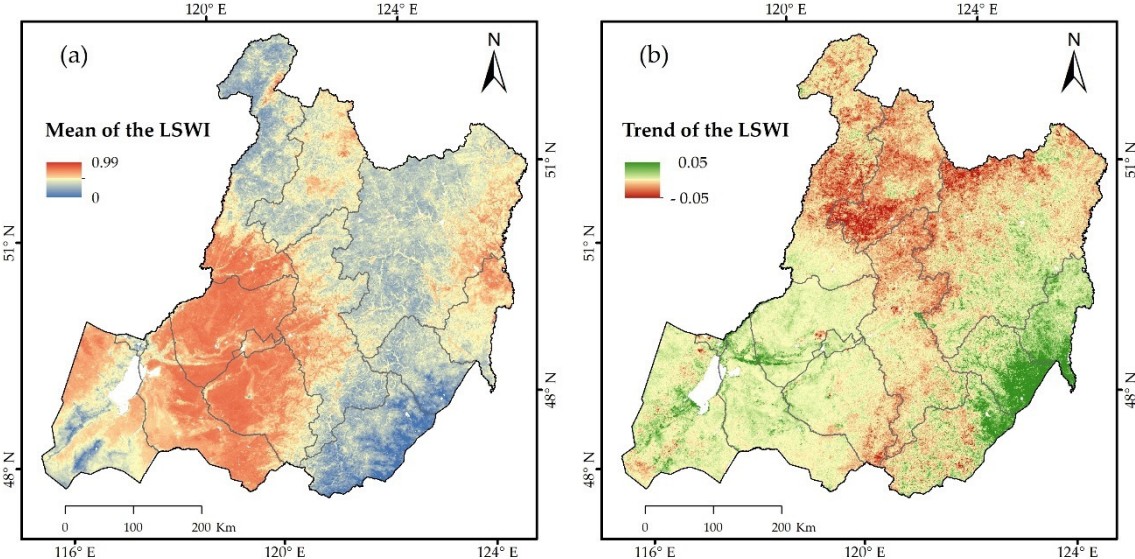

**Figure 6.** Distribution of (**a**) mean of Land Surface Water Index (LSWI) and (**b**) trend of LSWI in 2001–2018.

The mean LSTs in Hulunbuir exhibited a range of −1.89–23.55 °C. The values were low at high altitudes, i.e., the LSTs in the central region were lower than in the eastern and western parts. LSTs gradually increased from the middle to the boundaries of the study area, e.g., a low LST appeared in area b, while high LSTs occurred in areas e and j (Figure 7a). Over the past 18 years, the central and western regions of the study area mainly showed LSTs increasing, while a decrease was mainly distributed in the eastern part of the study area and area e, the west of area g, and the north of area j and areas f and i exhibited a stronger LST decrease (Figure 7b).

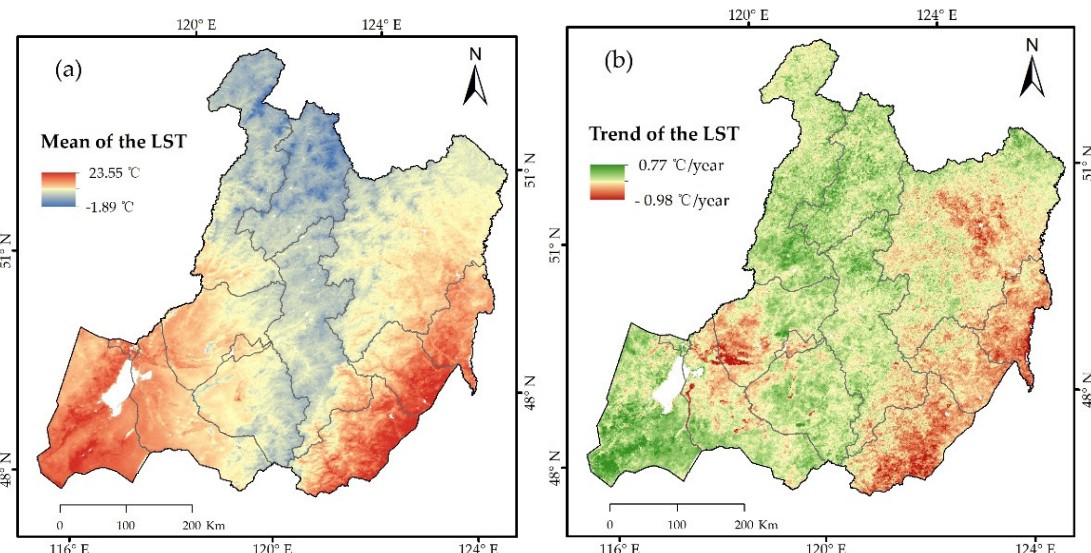

**Figure 7.** Distribution of (**a**) mean of land surface temperature (LST) and (**b**) trend of LST in 2001–2018.

### 3.2.2. Correlation between Snow Depth and LSWI

The melting of snow changes the surface humidity, and the land surface humidity affects the fuel humidity. Moreover, the fuel humidity is one of the important factors affecting wildfire occurrence. To further analyze the effects of snow depth on the SWF area in Hulunbuir, we calculated the spatial correlation coefficient between snow depth and the spring mean LSWI and LST over the period of 2001–2018. The spatial distribution of correlation coefficients and significance tests and between the two variables is shown in Figure 8.

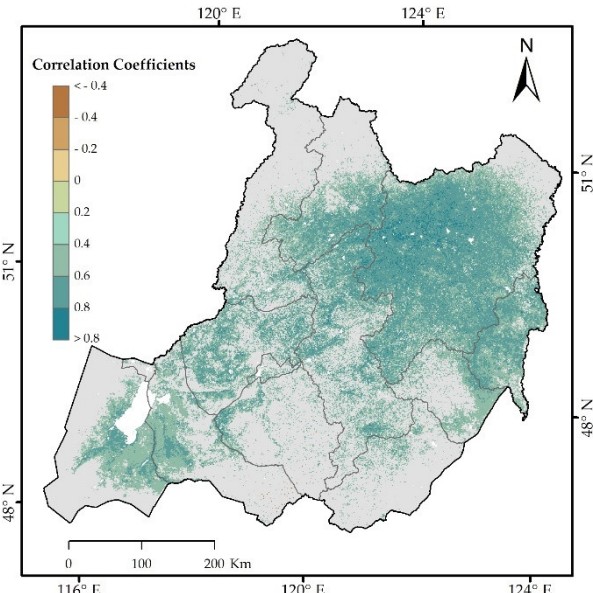

**Figure 8.** Correlation coefficients between snow depth and the LSWI during the period of 2001–2018.

Figure 8 shows that snow depth was positively correlated with the spring mean LSWI in most of Hulunbuir over the past 18 years, and the positive correlation was significant for 49.1% of total pixels. In areas c and f, the correlation was particularly obvious, and the highest correlation coefficient reached 0.85. Thus, increasing snow depth contributed to increased LSWI. However, the snow depth in the southern region of the study area was significantly negatively correlated with the LSWI, although less than 0.1% of the pixels were contained in this area. The grey area indicates no significant correlation in Figure 8.

### 3.2.3. Correlation between Snow Depth and LST

The melting of snow not only changes the surface humidity but also changes the LST. In this study, we calculated the spatial correlation coefficients between snow depth and the LST in the study area over the period of 2001–2018. The spatial distribution of the correlation coefficients between snow depth and the LST shows that the snow depth in the eastern and western of Hulunbuir was significantly negatively correlated with the LST, and the highest correlation coefficient reached −0.86 in area f (Figure 9). These results indicate that the increased snow depth contributed to a decrease in the LST. However, less than 1% of the study area showed significant negative correlations, and these points were scattered across area e. The grey area in Figure 9 indicates no significant correlation.

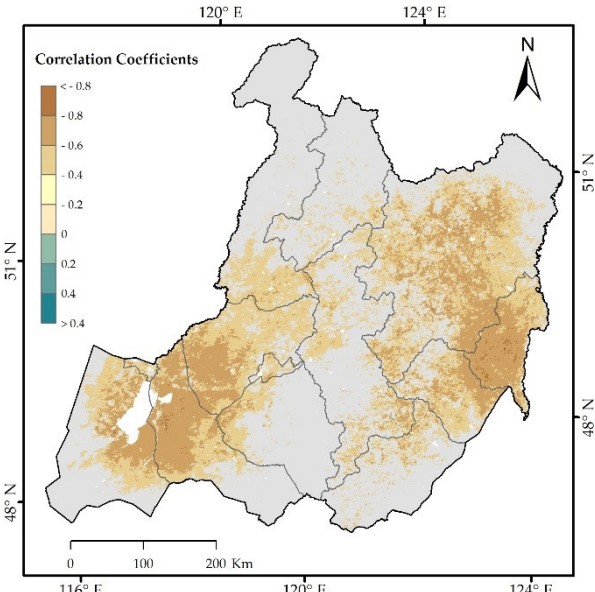

**Figure 9.** Correlation coefficients between snow depth and the LST during the period of 2001–2018.

*3.3. Future Prediction of Snow Depth in the Hulunbuir Region*

3.3.1. Future Snow Depth Based on the Hurst Exponent

The R/S analysis method was used to calculate the H (Hurst) exponent of snow depth for Hulunbuir during the period of 1979–2018 to predict future changes. The snow depth trend was determined from the 16 climate stations during the period of 1979–2018. The annual mean snow depth showed an increasing trend at the 16 climate stations in the study area during this period, and the H exponent of the annual mean snow depth was found to be 0.29 (Figure 10). Thus, in general, the trend in snow depth in the future is opposite the current state.

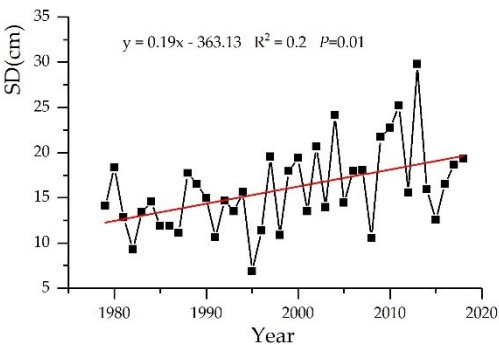

**Figure 10.** Interannual variation in mean snow depth during the period of 1979–2018.

In the past 40 years, snow depth increased in 54.9% of the area, with the highest trend being 0.79 cm/year; 45.1% of the area exhibited a decreasing trend, and these areas were mainly distributed in the northeastern part of the study area, with a few areas in the south and west (Figure 11). However, based on the R/S analysis of the 16 climate stations, the Hurst exponent of each station was less than 0.5 (Annex Table S1). Thus, the future trend in snow depth at each station is opposite the current state. This finding indicates that the trend of decreasing snow depth will switch in the future to an increasing trend. The area where snow depth will decrease in the future accounts for 54.9% of the total area.

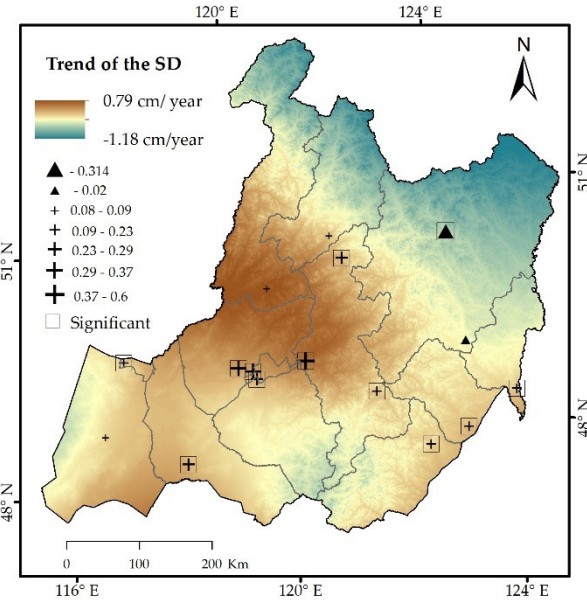

**Figure 11.** Spatial variation trend in annual snow depth in the Hulunbuir region during the period of 1979–2018 (+ indicates an increasing trend, ▲ indicates a decreasing trend, □ indicates a significant station).

### 3.3.2. Validation of Future Predictions Based on the Hurst Exponent

To validate the accuracy of using the Hurst exponent to predict the snow depth trend in the future, we validated the reliability of the H-based forecasting for the 16 climate stations. We divided the snow depth over the past 40 years into two periods, 1979–2012 and 2013–2018, and calculated their trends and Hurst exponents from 1979 to 2012 to predict the future annual snow depth change trend for the period of 2013–2018. The results are shown in Table 4 and Annex Figure S1.

**Table 4.** Trends and Hurst exponents of snow depth over the period of 1979–2012 and the forecasts and trends in snow depth over the period of 2013–2018.

| Station | 1979–2012 | | Hurst | | Forecast | 2013–2018 |
|---|---|---|---|---|---|---|
| | a | $p$ | H | $R^2$ | | a |
| 50425 | 0.190 | 0.100 | 0.354 | 0.726 | ↓ | −2.660 |
| 50431 | 0.220 | 0.060 | 0.317 | 0.800 | ↓ | −0.829 |
| 50434 | 0.440 | 0.001 | 0.401 | 0.902 | ↓ | −0.890 |
| 50445 | −0.400 | 0.050 | 0.451 | 0.931 | ↑ | 0.090 |
| 50514 | 0.240 | 0.003 | 0.429 | 0.892 | ↓ | −1.460 |
| 50524 | 0.390 | 0.002 | 0.318 | 0.604 | ↓ | −2.880 |
| 50525 | 0.260 | 0.009 | 0.390 | 0.870 | ↓ | −0.940 |
| 50526 | 0.760 | 0.000 | 0.413 | 0.888 | ↓ | −4.690 |
| 50527 | 0.370 | 0.002 | 0.320 | 0.814 | ↓ | 0.090 |
| 50548 | −0.040 | 0.720 | 0.244 | 0.434 | ↑ | −1.400 |
| 50603 | 0.100 | 0.100 | 0.301 | 0.681 | ↓ | −1.170 |
| 50618 | 0.200 | 0.030 | 0.258 | 0.610 | ↓ | 0.180 |
| 50632 | 0.250 | 0.040 | 0.471 | 0.944 | ↓ | −0.400 |
| 50639 | 0.080 | 0.300 | 0.339 | 0.882 | ↓ | 0.510 |
| 50645 | 0.110 | 0.200 | 0.328 | 0.752 | ↓ | −1.400 |
| 50647 | 0.200 | 0.040 | 0.392 | 0.830 | ↓ | −0.130 |

Note: a, $p$, H, $R^2$, ↑ and ↓ represent trend, significant level, Hurst index, goodness of fit, decreasing trend, and increasing trend, respectively.

For the study area, there were 14 climate stations with an increasing trend in snow depth from 1979 to 2012, of which 10 stations exhibited $p < 0.05$. However, the H exponents of these 14 stations

were all less than 0.5, and R$^2$ exceeded 0.6. These findings indicate that the snow depth at these stations will decrease in the future. The trend in snow depth during the period of 2013–2018 shows that the trend at 11 climate stations was consistent with the predicted results and exhibited a decreasing trend. For station 50445, the snow depth decreased from 1979 to 2012 and the H exponent was 0.451, indicating that the trend in snow depth in the future should be opposite to the current state. The validation results show that the trend in snow depth over the period of 2013–2018 was positive. Overall, 75% of the climate station snow depth predictions in the study area were accurate. Therefore, this study demonstrates that using the Hurst exponent to predict the future snow depth tendency is reliable.

## 4. Discussion

### 4.1. Variations in SWFs and Snow Depth

In the study area, there was considerable spatial heterogeneity between SWFs and snow depth. The results of this study indicate that SWFs are mainly distributed in eastern Hulunbuir, which is consistent with previous studies [8,36,38]. Liu et al. showed that grassland fires occur mainly in eastern Hulunbuir (with low occurrence in the other areas) based on the MODIS active fire product over the period of 2000–2014 [11]. This study showed that the land use patterns are quite different in the study area. The eastern land use type is mainly cropland, the central and northern areas are mainly forest, and the west is grassland and is sparsely populated. Li et al. reported that human activities are dense and the land use degree is high in the eastern region. Thus, fire occurrence in the eastern region is relatively high [52]. In addition, we showed that SWFs exhibited a decreasing trend over the past 18 years, and this was particularly large in 2003 and 2008.

Different from the spatial distribution of SWFs, snow depth in the middle of the study area was higher, while the east and west exhibited relatively low snow depths. The terrain of the study area is quite variable, and the central area includes the Greater Khingan Mountains. Because of its relatively high altitude and low temperature, the snow is not easily melted, which contributes to higher snow depth in the central part than in the east and west of the study area. In addition, we showed that snow depth exhibited a strong increasing trend over the period of 2001–2018. These conclusions are consistent with Wang et al., who, based on data at climate stations, studied the spatiotemporal variation characteristics of snow depth in the Hulunbuir region and found that snow depth increased during the period of 1960–2012 [53]. According to Zhong et al., the snow depth in the Hulunbuir area increased from 1966 to 2012 [54].

### 4.2. Relationships between SWFs and Snow Depth

The results showed a strong negative correlation between SWFs and snow depth. Previous research has shown that temperature, precipitation, and relative humidity are the main meteorological factors that affect wildfire occurrence [8,11,55]. Snow has a different influence on the surface humidity and the LST after it enters the melting period, and the surface humidity and temperature are the main climatic factors influencing wildfires. In this paper, we chose the LSWI and LST to evaluate the indirect impacts of snow depth on SWFs.

The results of this study showed that snow depth had a positive correlation with the spring LSWI and a negative correlation with the LST. That is, as snow depth increased, the LSWI also increased, and the LST showed a decreasing trend. This finding is similar to that of Liu et al., who found that snow in the permafrost regions of northeast China has a positive correlation with annual humidity and a negative correlation with temperature, which has a strong relationship with temperature [56]. We found that areas with higher snow depth had lower LSTs and higher LSWI values; however, SWFs were less likely to occur in these areas. In contrast, SWFs were more likely to occur in areas with lower snow depth. The LSWI in these places was relatively low, while the LST was higher. The LSWI in the eastern and western regions of the study area showed an increasing trend, while the SWF in the east

was more likely because the average LSWI in the area was lower. These findings extended those of Zhang et al. [12] and confirmed that the occurrence of SWFs was not only related to land use patterns and climatic factors during the fire occurrence but also closely related to winter snow. Our results provide an updated method and technical support for this type of wildfire disaster management. Additionally, these results expand the theory and methods of wildfire occurrence research and promote the research of disasters and disaster chains by enabling decision-makers to determine the possibility of wildfire in the spring of the next year according to the snow depth in the previous year and to prepare for disaster prevention.

### 4.3. Future Prediction of Snow Depth in the Hulunbuir Region

The results showed that snow depth in the study area exhibited an increasing trend over the past 40 years, and the results of the R/S analysis indicated that snow depth will decrease in the future. At the same time, we applied the mean snow depth data from 1979 to 2018. Morlet wavelet analysis showed that snow depth in the study area had obvious cyclical changes. After 1995, there was a relatively stable five-year cycle with a downward trend (Annex Figure S2). The results are consistent with the results of Wang et al. [53]. Furthermore, the prediction of snow depth at 16 climate stations in the study area was verified, and the prediction results of 12 climate stations were correct, indicating that it is feasible to use the Hurst exponent to predict the future trend in snow depth in Hulunbuir. In the future, the snow depth is expected to decrease across the Hulunbuir region. Therefore, wildfire prevention should be enhanced in the future spring seasons. In spring in Hulunbuir, the opening of fire prevention roads must be planned and fire prevention stations must be set up according to the previous year's snow depth conditions. In addition, to prevent wildfires, more fire prevention roads should be opened, more fire prevention stations should be set up, and more fire prevention materials should be stored in locations where SWFs are most likely to occur.

### 4.4. Uncertainty

Although our results are well correlated, certain influential factors are worth noting. For example, the topography is quite variable in the study area. The central part belongs to the Greater Khingan Mountains. Thus, the elevation is high and the temperature is relatively low. Therefore, snow is slower to melt after spring, which may lead to relatively large LSWI values and fewer wildfire occurrences. Moreover, it may be possible that human activities decrease as the slope increases. Hulunbuir is a region with high wildfire occurrence. Additionally, the snow depth data used in the methods of this paper were derived from discrete point data from climate stations. The advantage of this method is that it is simple and easy to implement, and the disadvantage is that the accuracy decreases as the distance from the observation site increases. Moreover, the interpolation method itself may impact the results. We used the Hurst exponent to predict the future snow depth trend in Hulunbuir. The disadvantage is that this analysis cannot predict how long the anticipated snow depth trend will continue in the future, and it is thus suitable for short-term trend prediction analysis. In future work, we should attempt to use the long-term series data for snow depth and SWFs to evaluate and analyze the influence of snow on wildfires in Hulunbuir. Additionally, we will consider more influential factors to make the analysis of the influencing factors of wildfires more comprehensive.

## 5. Conclusions

In this paper, we used SWFs extracted from the MCD64A1 dataset collected over the period of 2001–2018 and snow depth data to analyze the effect of snow on wildfire occurrence in Hulunbuir. The main influencing factors of LSWI and LST were selected to evaluate the indirect effect of snow depth on SWFs. The results provide a good reference for spring wildfire prevention.

Over the past 18 years, the area of SWFs gradually decreased, and there were more SWFs in 2003 and 2008. The frequency distribution and KDE results of SWFs showed that the main area of SWF occurrence was in the eastern part of the study area, and the SWFs in this area accounted for 88.59%

of all SWFs. In contrast to the spatial distribution of SWFs, snow depth in the middle of the study area was higher, while the east and west exhibited relatively low snow depths. Snow depth generally showed an increasing trend over the period of 2001–2018 and was mainly distributed in the east and west of the study area, while the snow depth in the Greater Khingan Mountains showed a decreasing trend. The temporal correlation coefficient between snow depth and SWFs was $-0.516$, while their spatial correlation coefficient was $-0.17$. Therefore, as the snow depth increases, the SWFs decrease, and the shallow snow depth area may have more SWF and vice versa.

Temperature and humidity are necessary external conditions for SWFs to occur. Our results also found that snow depth affected spring SWFs through influencing the spring LSWI and LST. The correlations between snow depth and the LSWI and snow depth and the LST showed that in regions where the snow depth was lower, the spring LSWI was lower and the LST was relatively high; these sites were more prone to SWFs and vice versa. In the future, the decreased snow depth will enhance the potential for SWFs in Hulunbuir. Thus, by using the snow information in the previous year, local fire officers may judge the possibility of SWFs in the next spring and make as much preparation as possible for the fire prevention.

**Supplementary Materials:** The following are available online at http://www.mdpi.com/2072-4292/11/3/321/s1, Figure S1: Trends of snow depth over the period of 1979–2012, 2013–2018 in the study area, Figure S2: 16 stations Morlet wavelet of snow depth in Hulnbuir during 1979–2018, Table S1: Trends and Hurst exponents of 16 climate stations snow depth over the period of 1979–2018 and the forecasts and trends in future.

**Author Contributions:** All authors contributed meaningfully to this study. Conceptualization, H.Z.; Methodology, H.Y.; Software, G.D., W.R.; Validation, Y.S., T.Y.; Formal Analysis, H.Y.; Data Curation, Y.S.; Writing—Original Draft Preparation, H.Y.; Writing—Review & Editing, H.Z.; Supervision, Y.S.

**Funding:** This work was supported by the National Natural Science Foundation of China (Grant numbers 41571489, 41871330, 41771450, 41501449 and 41601438) and Science and Technology Program Funding Project of Inner Mongolia Autonomous Region (Grant number 201502113).

**Conflicts of Interest:** The authors declare no conflict of interest.

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
