# Peer review of "The Effect of Snow Depth on Spring Wildfires on the Hulunbuir from 2001–2018 Based on MODIS"

_remotesensing, doi:10.3390/rs11030321_

Round 1

Reviewer 1 Report

The paper has been well prepared. But something still needs to be improved, English writing, abstract, conclusion and explanation of the physical mechanism. Please read my reviews and suggestions below. 

1.     Abstract is too detailed and includes too much unimportant information, such as correlation coefficients. Abstract should include major results summarized from these correlation coefficients and their implications for physical mechanism instead of the numbers.

2.     Line 16-19, please rewrite this sentence “In this study, …. In Hulunbuir.”  I think it will be better to describe like “In this study, based on MC464A1 burned area data and site snow depth measurements, multi diagnostic methods were used to evaluate the relationship between snow and spring wildfires (SWFs) in Hulunbuir. ” Lots of places in the text have similar English writing issues and need to be improved.

3.     Lines 40-41, wildfires also contribute to atmospheric pollution.

4.     Lines 42-43, I don’t understand “Wildfires can include grassland burning caused by natural or anthropogenic sources”, please explain more and add references if possible. 

5.     Line 43, replace “destructive power over” with “destroys on”.

6.     Line 48, “burning” should be “burned”.

7.     Based on the information from the context (i.e., lines 50-53), I doubt the so-called wildfires in your study probably are just agricultural fires which farmers burned the grass or leftover agricultural products for agricultural purpose. Please state it more clearly and provide corresponding evidences.

8.     Lines 75-83, there is something wrong about MODIS, for fire detection, the spatial resolution of the sensors on MODIS is 500 meters and measurements are twice per day (10:30 AM and 1:30 PM at day and 10:30 PM and 1:30 AM at night for Terra and Aqua respectively).

9.     Line 154, I think it is better to use “Meteorological Bureau” instead of “Weather Bureau”

10. Lines 168-174, this paragraph is useless for the paper, and I suggest to delete it.

11. Line 285, for the caption of Fig 3, I guess the word “frequency” (probably) should be added after (a).

12. Line 292, Figure 4b should be Figure 5b.

13. Line 312, the word meteorological is better to be replaced by weather or climate.

14. In Table 3, please make clear which aspect of SWF correlates with SD.

15. Labels for Figures 8 and 9, misleading information shows R<-0.8 insignificant in Figure 8 and R>0.5 insignificant in Figure 9.

16. In Section 3.3.2, line 390-405 for Table 4, please explain in detail how you did that, and why your method could get that change.  To me, it seems that the linear trend still keeps on rising, but a decadal change may make SD to be negative in the period of 2013-2018.

Author Response

Response to Reviewer 1 Comments

The paper has been well prepared. But something still needs to be improved, English writing, abstract, conclusion and explanation of the physical mechanism. Please read my reviews and suggestions below.

Your comment was highly insightful and enabled us to greatly improve the quality of our manuscript. In the following, we respond to your comments point by point.

Point 1: Abstract is too detailed and includes too much unimportant information, such as correlation coefficients. Abstract should include major results summarized from these correlation coefficients and their implications for physical mechanism instead of the numbers.

Response 1: Thank you for this suggestion. We rewrote the abstract and removed the unimportant information as per your suggestion.

Point 2:  Line 16-19, please rewrite this sentence “In this study, …. In Hulunbuir.”  I think it will be better to describe like “In this study, based on MC464A1 burned area data and site snow depth measurements, multi diagnostic methods were used to evaluate the relationship between snow and spring wildfires (SWFs) in Hulunbuir.” Lots of places in the text have similar English writing issues and need to be improved.

Response 2: Thank you for this suggestion. This sentence has been corrected in L15-L19 as follows:

“In this study, based on the Moderate Resolution Imaging Spectroradiometer (MODIS) burned area product MCD64A1 … in Hulunbuir.”

Point 3:   Lines 42-43, I don’t understand “Wildfires can include grassland burning caused by natural or anthropogenic sources”, please explain more and add references if possible.

Response 3: Thank you for this suggestion. The sentence is not appropriate in this article, and we have deleted it.

Point 4:   Lines 40-41, wildfires also contribute to atmospheric pollution.

Response 4: Thank you for this suggestion. This sentence has been corrected in L41 as follows:

“…and contribute to atmospheric pollution.”

Point 5:   Line 43, replace “destructive power over” with “destroys on”.

Response 5: Thank you for this suggestion. This sentence has been corrected in L41-L42 as follows: “Wildfires have considerable negative impacts on …”

 Point 6:   Line 48, “burning” should be “burned”.

Response 6: Thank you for this suggestion. This sentence has been corrected in L47 as follows:

 “… although the global burned …”

Point 7:   Based on the information from the context (i.e., lines 50-53), I doubt the so-called wildfires in your study probably are just agricultural fires which farmers burned the grass or leftover agricultural products for agricultural purpose. Please state it more clearly and provide corresponding evidences.

Response 7: Thank you for this suggestion. This sentence has been corrected in L49-L52 as follows:

The Inner Mongolia Autonomous Region of China (IMC) …human activities [8].

The refences are [7] and [8], and they have been added in L50 and L52, respectively.

Point 8:   Lines 75-83, there is something wrong about MODIS, for fire detection, the spatial resolution of the sensors on MODIS is 500 meters and measurements are twice per day (10:30 AM and 1:30 PM at day and 10:30 PM and 1:30 AM at night for Terra and Aqua respectively).

Response 8: Thank you for this suggestion. This sentence has been corrected in L74-L80 as follows:

“In addition, …, Terra and Aqua, respectively) [13].”

Point 9:   Line 154, I think it is better to use “Meteorological Bureau” instead of “Weather Bureau”

Response 9: Thank you for this suggestion. This sentence has been corrected in L152-L153 as follows:

“… acquired from the Meteorological Bureau of the IMC …”

Point 10:   Lines 168-174, this paragraph is useless for the paper, and I suggest to delete it.

Response 10: Thank you for this suggestion. We removed the paragraph as per your suggestion.

Point 11:   Line 285, for the caption of Fig 3, I guess the word “frequency” (probably) should be added after (a).

Response 11: Thank you for this suggestion. This sentence has been corrected in L282 as follows:

“The distribution frequency (a) and density (b) of SWFs during the period of 2001-2018.”

Point 12:   Line 292, Figure 4b should be Figure 5b.

Response 12: Thank you for this suggestion. This sentence has been corrected in L289 as follows:

“Fig. 5(b) shows the spatial variation in SD for Hulunbuir ...”

Point 13:   Line 312, the word meteorological is better to be replaced by weather or climate

Response 13: Thank you for this suggestion. This sentence has been corrected in L309 as follows:

“SWFs are affected by many external factors, such as climate conditions …”

Point 14:   In Table 3, please make clear which aspect of SWF correlates with SD.

Response 14: Thank you for this suggestion. This table3 has been corrected in L302 as follows:

“Spatial density of SWFs and spatial mean SD”

“Annual total SWF area and annual mean SD”

Point 15:   Labels for Figures 8 and 9, misleading information shows R<-0.8 insignificant in Figure 8 and R>0.5 insignificant in Figure 9.

Response 15: Thank you for this suggestion. We changed the legends in Figures 8 and 9.

Point 16:   In Section 3.3.2, line 390-405 for Table 4, please explain in detail how you did that, and why your method could get that change.  To me, it seems that the linear trend still keeps on rising, but a decadal change may make SD to be negative in the period of 2013-2018.

Response 16: The results presented in Section 3.3.1 indicated that the average SD of 16 sites showed an upward trend from 1979 to 2018 and the H index was 0.29, which indicated that in the future, the trend of SD will be opposite to the current state, that is, a downward trend will be observed. In Section 3.3.2, we validate whether the H index can effectively predict the change of SD. The verification step is as follows: We divided the 1979-2018 data for each of the Hulunbuir climate stations into two parts, 1979-2012 and 2013-2018. The trend analysis and H-index calculations were performed on the first part of the data, and the prediction results of the first part (the interannual changes of SD in Appendix Fig. S1 from 1979 to 2012 and 2013 to 2018) were validated with the second part of the data. The accuracy of the Hurst index prediction is 75%.

Reviewer 2 Report

The paper presented by the authors is interesting and has potentialities. However, there are some issues to be dealt with.

Abstract- MC464A1-What is this? From Modis? For the general public and a wider scientific audience, you must use clearer language.

The abstract could be shortened and more direct. The abstract is confusing. You describe, with excessive detail, the results. Please rewrite the abstract.

L54-Hulunbui-Here you should explain which part of Inner Mongolia Autonomous Region of China regards to Hulunbui and what it is (subregion? Administrative region?). We only know that when you present in detail the study area after the introduction.

L63- “Many researchers”-Please provide examples.

L62-L103- This paragraph is too long. Consider separating it in several paragraphs, according to the ideas presented.

L119- “The study area is located in the city of Hulunbuir”- Or in a administrative region?

L159-166- This explanation should be inserted after ANUSPLIN 4.3, without inserting other ideas.

Methods-The methods are described satisfactorily and may be reproducible. The meaning of each equation is clear.

Line 207- “d>0”-If you are describing the meaning of d, it should be referring to it iniatilly only (consider writing only the variable letter “d”).

“3.3.2. Validation of Future Predictions Based on the Hurst Exponent”-The validation process seems OK.

Discussion-It is satisfactorily presented.

You mention  Line 384 -Annex Table S1 and in line 489 Annex Figure S2, despite not existing annexes to this paper. 

Author Response

Response to Reviewer 2 Comments

The paper presented by the authors is interesting and has potentialities. However, there are some issues to be dealt with.

Thank you very much for your comments. Your comments have greatly improved our manuscript. We respond to your comments point by point below.

Point 1: Abstract - MC464A1-What is this? From Modis? For the general public and a wider scientific audience, you must use clearer language.

Response 1: Thank you for this suggestion. This sentence has been corrected in L16-L17 as follows:

“… based on the Moderate Resolution Imaging Spectroradiometer (MODIS) burned area product MCD64A1…”

Point 2:  The abstract could be shortened and more direct. The abstract is confusing. You describe, with excessive detail, the results. Please rewrite the abstract.

Response 2: Thank you for this suggestion. We rewrote the abstract as per your suggestion.

Point 3:    L54-Hulunbui-Here you should explain which part of Inner Mongolia Autonomous Region of China regards to Hulunbui and what it is (subregion? Administrative region?). We only know that when you present in detail the study area after the introduction.

Response 3: Thank you for this suggestion. This sentence has been corrected in L53-L54 as follows:

“Hulunbuir is the easternmost prefecture-level city in IMC, and it is rich …”

Point 4:   L63- “Many researchers”-Please provide examples.

Response 4: Thank you for this suggestion. The references are [14-16], which have been added in L64.

Point 5:   L62-L103- This paragraph is too long. Consider separating it in several paragraphs, according to the ideas presented.

Response 5: Thank you for this suggestion. We divide the paragraph into two endings starting at L62 and L99.

Point 6:   L119- “The study area is located in the city of Hulunbuir”- Or in a administrative region?

Response 6: Thank you for this suggestion. This sentence has been corrected in L117 as follows:

“The study area is located in the Hulunbuir (prefecture-level city) in the IMC …”

Point 7:   L159-166- This explanation should be inserted after ANUSPLIN 4.3, without inserting other ideas.

Response 7: Thank you for this suggestion. This explanation has been corrected in L157 as follows:

ANUSPLIN 4.3, which is a special software package for surface fitting of climate data.”

Point 8:   Line 207- “d>0”-If you are describing the meaning of d, it should be referring to it iniatilly only (consider writing only the variable letter “d”).

Response 8: Thank you for this suggestion. This sentence has been corrected in L201 as follows:

“… d is the bandwidth used …”

Point 9:   You mention Line 384 -Annex Table S1 and in line 489 Annex Figure S2, despite not existing annexes to this paper.

Response 9: The annexes were uploaded as a compressed file, so it is unclear why you did not receive the annexes. We will gladly send you another copy if required.

Round 2

Reviewer 1 Report

This revised manuscript has been improved since its first version, however, its English writing still has issues. Besides the issues that I found below, I strongly suggest the authors should make another big effort to improve it further. Thanks.

1.     Lines 14 and 59, it is better to use the word “detailed” instead of “accurate”.

2.     Line 28, replace “found” with “indicated” or “revealed”.

3.     Line 41, add words “as well” after “pollution”.

4.     Line 57, change “ gale force wind” with “gale” or “strong wind”. 

5.     Lines 63 and following text, too many abbreviation words in this paper which cause difficulties to read. For example, for remote sensing, I don’t think it is necessary to use RS. SD (snow depth) is another example. Please search on the web for when, where or how to use the initial abbreviation.

6.     Line 125, replace “precipitation” with the phase “the annual precipitation”.

7.     Line 313, remove “of” in the phase “In February of each year”.

8.     Line 314, replace “and SD decrease” with “of snow depth decrease”; change “The melting snow” with “Snow melting”.

9.     Line 333, change “showed increased LSTs” to “showed LSTs increasing”.

10. Line 334, west of g --> the west of Area g; north of j --> the north of Area j.

11. Lines 337-338, it is better to change the caption of Figure 6 to “Distributions of (a) mean of LSWI and (b) trend of LSWI in 2001-2018.”. The same revision should apply to the caption of Figure 7 but for LST.

12. Line 348, remove “using R”. The following “R” should be replaced by correlation coefficient, and “P” should be changed by significant test.

13. Line 402, opposite --> opposite to.

14. Line 423, “resulting in SDs in the central part of the study area being higher than in the east and west” --> “ which contributes to higher snow depth in the central part than in the east and west of the study area”.

15. Line 425, delete “the smallest being in 2008”.

16. Line 489, “The results of this study” --> “The results”.

17. Lines 497-499, If it is for me, I would like to rewrite this part as “The temporal  correlation coefficient between snow depth and SWFs is −0.516, while their spatial correlation coefficient is −0.17. That means, as the snow depth increases, the SWFs decrease, and the shallow snow depth area may have more SWFs , and vice versa.

18. Line 501, “by” --> “through”.

19. Lines 504-506, you can improve this part as “In the future, the snow depth decrease will enhance the potential more SWFs in Hulunbuir. Thus, by using the snow information in the previous year, the local fire officers may judge the possibility of SWFs in the next spring and make preparation as much as possible for the fire prevention.”.

Author Response

Response to Reviewer 1 Comments

This revised manuscript has been improved since its first version, however, its English writing still has issues. Besides the issues that I found below, I strongly suggest the authors should make another big effort to improve it further. Thanks.

Thank you very much for your comments. Your comments have greatly improved our manuscript. We respond to your comments point by point below.

Point 1: Lines 14 and 59, it is better to use the word “detailed” instead of “accurate”.

Response 1: Thank you for this suggestion. This sentence has been corrected in L14 and L59 as follows:

      L14: Detailed research on the impact of wildfires …”

      L59: Therefore, detailed research on the …”

Point 2:  Line 28, replace “found” with “indicated” or “revealed”.

Response 2: Thank you for this suggestion. This sentence has been corrected in L28 as follows:

“The results of the Hurst exponent indicated that …”

Point 3:   Line 41, add words “as well” after “pollution”.

Response 3: Thank you for this suggestion. This sentence has been corrected in L41 as follows:

; and they contribute to atmospheric pollution as well.”

Point 4:   Line 57, change “gale force wind” with “gale” or “strong wind”.

Response 4: Thank you for this suggestion. This sentence has been corrected in L57 as follows:

“…, and a high number of strong wind days.”

Point 5:   Lines 63 and following text, too many abbreviation words in this paper which cause difficulties to read. For example, for remote sensing, I don’t think it is necessary to use RS. SD (snow depth) is another example. Please search on the web for when, where or how to use the initial abbreviation.

Response 5: Thank you for this suggestion. We changed the RS to remote sensing and changed the SD in the full text to snow depth.

 Point 6:    Line 125, replace “precipitation” with the phase “the annual precipitation”.

Response 6: Thank you for this suggestion. This sentence has been corrected in L125-L126 as follows:

 “…, and the annual precipitation is …”

Point 7:   Line 315, remove “of” in the phase “In February of each year”.

Response 7: Thank you for this suggestion. This sentence has been corrected in L315 as follows:

“… gradually beginning in February each year, …

Point 8:  Line 314, replace “and SD decrease” with “of snow depth decrease”; change “The melting snow” with “Snow melting”.

Response 8: Thank you for this suggestion. This sentence has been corrected in L315-L316 as follows:

“…, and the coverage of snow depth decrease gradually. Snow melting will affect…”

Point 9:  Line 333, change “showed increased LSTs” to “showed LSTs increasing”.

Response 9: Thank you for this suggestion. This sentence has been corrected in L336 as follows:

“… the study area mainly showed LSTs increasing, …”

Point 10:  Line 334, west of g --> the west of Area g; north of j --> the north of Area j.

Response 10: Thank you for this suggestion. This sentence has been corrected in L337 as follows:

“…, the west of area g, and the north of area j, …”

Point 11:  Lines 337-338, it is better to change the caption of Figure 6 to “Distributions of (a) mean of LSWI and (b) trend of LSWI in 2001-2018.”. The same revision should apply to the caption of Figure 7 but for LST.

Response 11: Thank you for this suggestion. This sentence has been corrected in L299, L340 and L343 as follows:

L299: “Figure 5. Distribution of (a) mean of snow depth and (b) trend of annual snow depth in 2001–2018.”

L340: “Figure 6. Distribution of (a) mean of LSWI and (b) trend of LSWI in 2001–2018.”

L343: “Figure 7. Distribution of (a) mean of LST and (b) trend of LST in 2001–2018.”

Point 12:  Line 348, remove “using R”. The following “R” should be replaced by correlation coefficient, and “P” should be changed by significant test.

Response 12: Thank you for this suggestion. This sentence has been corrected in L349 as follows:

“… period of 2001–2018. The spatial distribution of correlation coefficients and significant test ...”

Point 13:   Line 402, opposite --> opposite to.

Response 13: Thank you for this suggestion. This sentence has been corrected in L404 as follows:

“… in the future should be opposite to the current state.”

Point 14:  Line 423, “resulting in SDs in the central part of the study area being higher than in the east and west” --> “ which contributes to higher snow depth in the central part than in the east and west of the study area”.

Response 14: Thank you for this suggestion. This sentence has been corrected in L426-L427 as follows:

“…, which contributes to higher snow depth in the central part than in the east and west of the study area.”

Point 15:   Line 425, delete “the smallest being in 2008”.

Response 15: Thank you for this suggestion. This sentence has been corrected in L428 as follows:

“…, snow depth exhibited a strong increasing trend over the period of 2001–2018, …”

Point 16:  Line 489, “The results of this study” --> “The results”.

Response 16: Thank you for this suggestion. This sentence has been corrected in L493 as follows:

“The results provide a good reference for spring wildfire prevention.”

Point 17:  Lines 497-499, If it is for me, I would like to rewrite this part as “The temporal correlation coefficient between snow depth and SWFs is −0.516, while their spatial correlation coefficient is −0.17. That means, as the snow depth increases, the SWFs decrease, and the shallow snow depth area may have more SWFs, and vice versa.”

Response 17: Thank you for this suggestion. This sentence has been corrected in L501-L503 as follows:

“The temporal correlation coefficient …, and vice versa.”

Point 18:   Line 501, “by” --> “through”.

Response 18: Thank you for this suggestion. This sentence has been corrected in L505 as follows:

“… snow depth affected spring SWFs through influencing the spring LSWI and LST.”

Point 19:   Lines 504-506, you can improve this part as “In the future, the snow depth decrease will enhance the potential more SWFs in Hulunbuir. Thus, by using the snow information in the previous year, the local fire officers may judge the possibility of SWFs in the next spring and make preparation as much as possible for the fire prevention.”.

Response 19: Thank you for this suggestion. This sentence has been corrected in L508-L511 as follows:

“In the future, … as much as possible for the fire prevention.”
